# Telomere Shortening in Three Diabetes Mellitus Types in a Mexican Sample

**DOI:** 10.3390/biomedicines11030730

**Published:** 2023-02-28

**Authors:** Pavel Cuevas Diaz, Humberto Nicolini, German Alberto Nolasco-Rosales, Isela Juarez Rojop, Carlos Alfonso Tovilla-Zarate, Ester Rodriguez Sanchez, Alma Delia Genis-Mendoza

**Affiliations:** 1Genomics of Psychiatric and Neurodegenerative Diseases Laboratory, National Institute of Genomic Medicine (INMEGEN), Mexico City 01090, Mexico; 2Biomedical Postgraduate Program, Academic Division of Health Sciences, Juarez Autonomous University of Tabasco, Villahermosa 86000, Mexico; 3Academic Division of Health Sciences, Juarez Autonomous University of Tabasco, Villahermosa 86000, Mexico; 4Comalcalco Multidisciplinary Academic Division, Juarez Autonomous University of Tabasco, Comalcalco 86040, Mexico; 5Diabetes Clinic, Hospital Regional de Alta Especialidad “Dr. Gustavo A. Roviosa Perez”, Villahermosa 86020, Mexico; 6Psychiatric Attention Services, Hospital Psiquiatrico Infantil, “Dr. Juan N. Navarro, Tlalpan”, Mexico City 14080, Mexico

**Keywords:** telomere length, telomere shortening, diabetes, Mexicans

## Abstract

This study aimed to explore the role of telomere length in three different diabetes types: latent autoimmune diabetes of adulthood (LADA), latent autoimmune diabetes in the young (LADY), and type 2 diabetes mellitus (T2DM). A total of 115 patients were included, 72 (62.61%) had LADA, 30 (26.09%) had T2DM, and 13 (11.30%) had LADY. Telomere length was measured using real-time Polymerase Chain Reaction. For statistical analysis, we used the ANOVA test, X2 test, and the Mann–Whitney U test. Patients with T2DM had higher BMI compared to LADA and LADY groups, with a BMI average of 31.32 kg/m^2^ (*p* = 0.0235). While the LADA group had more patients with comorbidities, there was not a statistically significant difference (*p* = 0.3164, *p* = 0.3315, *p* = 0.3742 for each of the previously mentioned conditions). There was a difference between those patients with T2DM who took metformin plus any other oral antidiabetic agent and those who took metformin plus insulin, the ones who had longer telomeres. LADA patients had shorter telomeres compared to T2DM patients but not LADY patients. Furthermore, T2DM may have longer telomeres thanks to the protective effects of both metformin and insulin, despite the higher BMI in this group.

## 1. Introduction

Diabetes mellitus (DM) is a metabolic disease characterized by chronic hyperglycemia due to a lack of insulin secretion, insulin sensitivity/action, or both, thus requiring frequent monitoring and proper control with both lifestyle changes and pharmacotherapy [1,2]. Diabetes mellitus can be classified into various types; however, the most common types are type 1 (T1DM) and type 2 (T2DM). Generally speaking, T1DM is due to the destruction of pancreatic β-cells by T-cell-mediated autoimmunity, usually leading to absolute insulin deficiency; this category includes latent autoimmune diabetes of adulthood (LADA) and latent autoimmune diabetes in the young (LADY). Otherwise, T2DM is due to a progressive loss of adequate β-cell insulin secretion, usually in the background of some degree of insulin resistance. Despite classification being important, DM is a heterogeneous disease, in which clinical presentation and progression may vary considerably among individuals, leading to some patients not being clearly classified as having T1DM or T2DM at the time of diagnosis [3].

Mexico is one of the leading countries in the incidence of diabetes worldwide. According to the International Diabetes Federation Atlas from 2021 [4], diabetes prevalence in Mexican adults is 16.9% (14.2 to 22.1, 95% CI), with a possible increase to 18.3% by 2030, meaning 17,062.7 in 100,000 s Mexicans with diabetes. Considering that T1DM and T2DM are also polygenic diseases [5], the heritability is 15 for T1DM and 3 for T2DM [6], resulting in a lifetime risk of 5% of developing T1DM if a parent has type 1 diabetes, higher if the father has the disease [7], and 40% T2DM if one parent has type 2 diabetes, higher if the mother has the disease [8]. One of the genetic compounds included in DM pathophysiology is telomere length, which is found to be heritable [9].

Telomeres are double-stranded DNA protein regions with a variable length located at the end of all chromosomes, except for the very end of the strand, which is single-stranded [10,11]. Telomeres are composed of several repeats of the hexanucleotide TTAGGG, and thus are rich in guanine [12]. This sequence maintains the stability of chromosomes, prevents the end fusion of chromosomes, protects chromosome structure, and determines the lifespan of cells [13]. To maintain the length of the telomeres, there is a ribonucleoprotein named Telomerase, which has a reverse transcriptase activity, adding telomeric DNA repeats to the single-strand overhang of telomeres, playing a crucial role in cell proliferation, differentiation, and survival [14,15]. Experimental and clinical evidence indicates that the lymphocyte telomere length corresponds to the telomere length of the stem cells and endothelial progenitor cells. Therefore, the lymphocyte length in studies, could be used as a biomarker of cell ageing [16].

The first team to demonstrate an association between type 2 diabetes mellitus (T2DM) and shortened telomere length were Jeanclos et al. back in 1998 [17]. Since then, numerous studies have shown a relationship between telomere attrition and diabetes, such as the meta-analysis of prospective, case-control, and cross-sectional studies performed by Zhao et al. which included nine population cohorts, 5759 cases, and 6518 controls, and demonstrated that leukocyte telomere length is associated with T2DM risk [18].

One of the possible mechanisms by which DM can cause telomere shortening is oxidative stress. It is well established that hyperglycemia elicits an increase in reactive oxygen species production and triggers maladaptive response by affecting several metabolic and signaling pathways leading to DNA damage, such as single-strand breaks and telomere erosion [19]. Short telomeres may lead to premature β-cell senescence, resulting in reduced β-cell mass and, subsequently, impaired insulin secretion and glucose tolerance [20]. Oxidative stress induces an abnormal telomere–telomerase system function, leading to DNA damage and b-cell dysfunction, resulting in further aggravation or development of diabetes mellitus, being a cyclical process [11,13,21,22].

A 2016 meta-analysis found telomere length variation by geographical region, European patients displaying a reduced telomere length compared with Asian and American populations. In addition, patients with T1DM were found to have a shorter telomere length compared to patients with T2DM, and females had shorter telomeres than males [23]. However, other published studies have demonstrated the opposite [24,25]. However, little is known about the relationship between telomere length and the Mexican population. Furthermore, we are still unfamiliar with the link between telomere length and different diabetes classifications among Mexicans. Therefore, the aim of this study was to investigate the relationship between telomere shortening and different types of diabetes in a Mexican population sample.

## 2. Materials and Methods

### 2.1. Patients Data

We performed a cross-sectional study with samples obtained from Mexican patients attending an external diabetes consult in the Hospital Regional de Alta Especialidad Dr. Gustavo A. Rovirosa Pérez. A total of 200 patients with diabetes were initially considered for this study, but after adjusting them to the inclusion and exclusion criteria mentioned later, only 115 were finally included in this study. Patients were previously evaluated and diagnosed by an endocrinologist according to the American Diabetes Association criteria 2021. From the total of patients, 72 (62.61%) had latent autoimmune diabetes of adulthood (LADA), 30 (26.09%) had type 2 diabetes mellitus (T2DM), and 13 (11.30%) had latent autoimmune diabetes in the young (LADY). The mean age was 55.30 years, 71.30% (*n* = 82) being women and 28.70% (*n* = 33) men.

The inclusion criteria were: (a) patients previously diagnosed with type 1 or type 2 diabetes mellitus, (b) Mexican patients with both Mexican parents and grandparents, and (c) patients that agree to be part of the study. The exclusion criteria were: (a) foreign patients and (b) patients who did not accept the blood sample collection. During the data collection, we also registered socioeconomic data (age, sex, occupation, educational status, and civil status), clinical characteristics, including other comorbidities and diabetes complications, alcohol consumption, smoking status, and pharmacologic treatment data.

In Table 1, we show the values for Glutamic Acid Decarboxylase Antibodies (GADA), which were measured in all patients. GADA were used to determine the type of diabetes of each patient with the use of the Human Anti-Glutamic Acid Decarboxylase Antibodies ELISA Kit, following the manufacturer’s instructions. Patients with GADA > 5 IU/mL and >30 years old were diagnosed as LADA, those with GADA > 5 IU/mL and <30 years old were diagnosed as LADY, and those with GADA < 5 IU/mL as T2DM.

### 2.2. Anthropometric and Laboratory Assessment

At the time of the study, the mean diagnosis time was 16.02 years, and at physical exploration, there was a mean body mass index (BMI) of 30.26 kg/m^2^. At the time of blood sample extraction, biochemical tests were performed, resulting in a mean glycated hemoglobin (HbA1c) of 6.48% and a mean fasting plasma glucose of 133.03 mg/dL. Out of the total of patients, 52 (45.22%) had hypertension, 28 (24.35%) had dyslipidemia, and 5 (4.35%) had neuropathy, the LADY population having the most comorbidities (Table 1). Among the antidiabetic agents, 110 (95.65%) used metformin, 45 (39.13%) used glibenclamide, 29 (25.22%) used any dipeptidyl peptidase 4 inhibitors, and 84 (73.04%) used any kind of insulin (Table 2). Other laboratory parameters were assessed (high-density lipoprotein cholesterol, low-density lipoprotein cholesterol, triglycerides, insulin and glomerular filtration rate, human islet antigen-2 antibody, and human zinc transporter8 antibody) but were not taken into consideration for this study as they were not relevant to the study.

### 2.3. Measurement of Telomere Length

First, blood samples were obtained from all patients in the Diabetes Clinic of the Regional Hospital of High Specialty “Dr. Gustavo A. Rovirosa Pérez” in Hospital Re-gional de Alta Especialidad “Dr. Gustavo A. Rovirosa Pérez”. After getting one 4 mL blood tube of each patient, the samples were sent to the National Institute of Genomic Medicine in Mexico City.

DNA extraction was obtained from peripheral leukocytes using the Gentra Puregene (Qiagen) commercial method. Quantity and quality were assessed using spectrophotometry (Nanodrop 2000). To determine telomere length, we performed a real-time Polymerase Chain Reaction (rt-PCR), along with an SYBR Green master mix. To obtain the relative telomere to single copy gene (T/S) ratio, sample T/S was divided by reference T/S, using Succinate dehydrogenase complex flavoprotein subunit A (SDHA) as the gene control. The primers were as follows:Telomere-F 5′-CGG TTT GTT TGG GTT TGG GTT TGG GTT TGG GTT TGG GTT-3′Telomere-R 5′-GGC TTG CCT TAC CCT TAC CCT TAC CCT TAC CCT TAC CCT-3′SDHA-F 5′-TCT CCA GTG GCC AAC AGT GTT-3′SDHA-R 5′-GCC CTC TTG TTC CCA TCA AC-3′

### 2.4. Statistical Analysis

The values of continuous variables between groups were compared using the ANOVA test, and the categorical variables were compared using an X2 test. The Mann–Whitney U test was performed if the data had skewed distributions. To achieve an approximately normal distribution, log transformation for telomere length was used. All statistical analysis and graphics were created using GraphPad Prism 8.0 for Windows, setting *p* < 0.05 as statistically significant.

## 3. Results

A total of 115 patients with any diabetes classification were included in this study, only divided into latent autoimmune diabetes of adulthood (LADA), latent autoimmune diabetes in the young (LADY), and type 2 diabetes mellitus (T2DM). Each group had its distinctive characteristics. Patients with T2DM had higher BMI compared to LADA and LADY groups, with a BMI average of 31.32 kg/m^2^ (*p* = 0.0235). On the other hand, the T2DM group had more glucose control, with a mean HbA1c of 5.28%. However, these differences among the three groups were not statistically significant (*p* = 0.8736).

It is worth noting that all three groups had patients with comorbidities. In this study, we only included systemic arterial hypertension, dyslipidemia, and neuropathy. While the LADA group had more patients with comorbidities, there was not a statistically significant difference (*p* = 0.3164, *p* = 0.3315, *p* = 0.3742 for each of the previously mentioned conditions).

A correlation analysis between telomere length and three variables, age, BMI, and HbA1c, was performed to explore which factors affected telomere length. None of the variables were significantly associated with telomere length (Table 3, Figure 1). However, in the LADA group, both age and age of disease diagnosis were correlated with telomere length, with a *p* < 0.0001 using the Mann–Whitney test (Table 4).

In the first sight, the telomere length of the LADA group was significantly shorter than those of the T2DM and LADY group. However, after performing the Mann–Whitney test, only the LADA group had significantly shorter telomeres than those of the T2DM group (*p* = 0.0121, Figure 2a), while it was not significant against those with LADY (*p* = 0.8221, Figure 2b). Between T2DM and LADY groups, there was no statistical difference (*p* = 0.1671, Figure 2c). After comparing the three groups simultaneously, there was a significant difference in telomere lengths (*p* = 0.0424, Figure 2d).

As all individuals were taking at least two antidiabetic agents, we explored the relationship between metformin plus any other oral antidiabetic agent (OAA) and metformin plus insulin. There was a difference between those patients with T2DM who took metformin plus any other OAA and those who took metformin plus insulin, the ones who had longer telomeres (*p* = 0.0256, Figure 3b).

Regardless of the diabetes group, there was not any difference between metformin plus any other OAA and metformin plus insulin (*p* = 0.1478). When dividing into subgroups, no analysis was performed in the LADY group, as all of them except for one were taking insulin. As for the LADA group, there was no difference (*p* = 0.3759, Figure 3a).

## 4. Discussion

As previously mentioned, diabetes is a heterogeneous disease affecting the entire globe, and is prevalent in Mexico. A meta-analysis of 17 papers conducted by Wang et al. determined that telomere length in diabetic patients varied based on geographical region. This study found that European patients with diabetes displayed a more pronounced telomere length compared with Asian and US patients. Additionally they demonstrated that telomere length was affected by diabetes type, BMI, age and sex [26]. These results agree with other meta-analyses that concluded that a significant association between diabetes and telomere length was influenced by geographical region and diabetes type [18,27]. Additionally, telomere shortening is widely accepted as a hallmark of cellular senescence. In our study, LADA patients were found to have shorter telomere lengths compared with T2DM patients, and no difference when compared to LADY patients. The previous results were independent of age, BMI, HbA1c, and comorbidities, as these were not significant. BMI was the only significant clinical variable, mainly in the T2DM group, which was found to have a higher body mass index in comparison to the LADA and LADY groups. BMI and sex have been described as strong predictors of the association between diabetes and telomere length [26]. This is supported by a study conducted by Al-Thuwaini, who found that higher waist circumference, along with higher HbA1c levels and fasting blood glucose and lower high-density lipoprotein to be correlated with shortened telomere length in diabetic patients [28]. Other studies have supported these conclusions [29]. In fact, patients with T2DM who took metformin and insulin had longer telomere lengths. This is supported by previously published evidence, suggesting that insulin and metformin have protective effects [30,31]. Metformin has been found to delay the senescence of renal tubular epithelial cells in diabetic nephropathy [32] and reduce inflammation [33], irrespective of diabetes status, and alleviate the effects of aging on diabetic progression [34,35]. Even exposure to metformin and insulin can prevent telomere attrition in individuals with prediabetes [36,37]. A profound review on molecular mechanisms can be found in Kulkarni et al. [38], Hu et al., and Mohammed et al. [39].

Not only metformin appears to have a beneficial effect on telomere length. Other antidiabetic agents such as glibenclamide and sitagliptin have also been studied. Glibenclamide has been found to remarkably decrease the telomere shortening rate. This may be due to the fact that glibenclamide stimulates insulin secretion from pancreatic β-cells and reduces free fatty acid concentrations [40]. A randomized controlled trial reported a significant improvement in glucose control, reduced insulin resistance, and elongated telomere length in T2DM patients treated with sitagliptin for 2 months. It was also reported to have no effect on telomerase activity between patients and control groups [41].

Insulin and other antidiabetic agents have been found to possess anti-inflammatory effects. This can be due to reductions in nicotinamide adenine dinucleotide phosphate hydrogen (NADPH) oxidase expression, reactive oxygen species generation, and NF-κB binding [42], in the case of insulin. On the other hand, Zeng et al. found that insulin therapy may accelerate telomere shortening. It is possible that this may be due to the association between insulin and weight gain, aggravating the insulin resistance, promoting oxidative stress, and shortening the telomere length [43].

The advantage of our study is mainly being the first population-based study comparing three different diabetes types. First, the LADA group was demonstrated to have a shorter telomere length than the T2DM group. Second, our study confirmed that metformin plus insulin may have a role preventing shortening in the T2DM group.

We must not forget the limitations of this study. First, this is a cross-sectional study; thus, speculating the causal relationship between telomere lengths in these diabetes groups is difficult. Second, the number of subjects was not very large, considering the overall population. Finally, subjects were obtained from a specialized diabetes clinic, meaning that patients in our study had access to an antibody-detecting facility, nutrition counseling, and constant check-ups; most of the patients were under stricter control than those who diabetes can be found in frequently.

## 5. Conclusions

A great variety of molecular pathways have been sketched in an attempt to elucidate the mechanisms by which diabetes and telomere attrition are linked. Discovering new pathways could be useful as potential therapy targets and even more personalized treatments. All available antidiabetics have shown some degree of protection against telomere shortening in diabetic patients. Epigenetic variables may also contribute to glycemic and metabolic control in order to make a synergic effort to minimize cellular damage. It is evident that further studies will be needed to analyze the possible variables that could be biased, including more gender, age, and nationality studies.

This study indicated that LADA patients had shorter telomeres compared to T2DM patients but not LADY patients, and T2DM may have longer telomeres thanks to the protective effects of both metformin and insulin, despite the higher BMI in this group. To fully elucidate these results, more research is needed in the future with more subjects, taking into consideration the antibodies testing to fully clarify the proper diabetes mellitus groups.

## Figures and Tables

**Figure 1 biomedicines-11-00730-f001:**
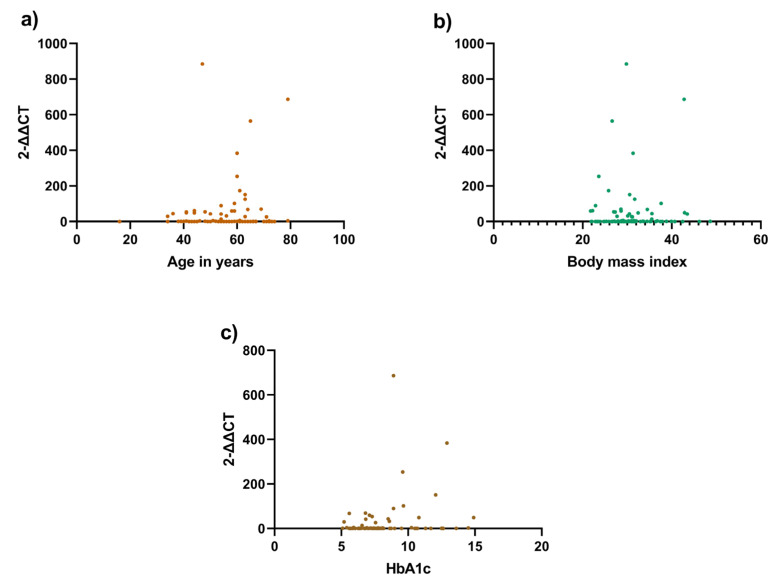
Correlation analysis between telomere length and clinical variables. Correlation between the telomere length using rt-PCR and clinical variables: (**a**) Telomere length and age (*p* = 0.9004); (**b**) Telomere length and body mass index (*p* = 0.1748); (**c**) Telomere length and HbA1c (*p* = 0.3335). All three variables were found not to be statistically significant.

**Figure 2 biomedicines-11-00730-f002:**
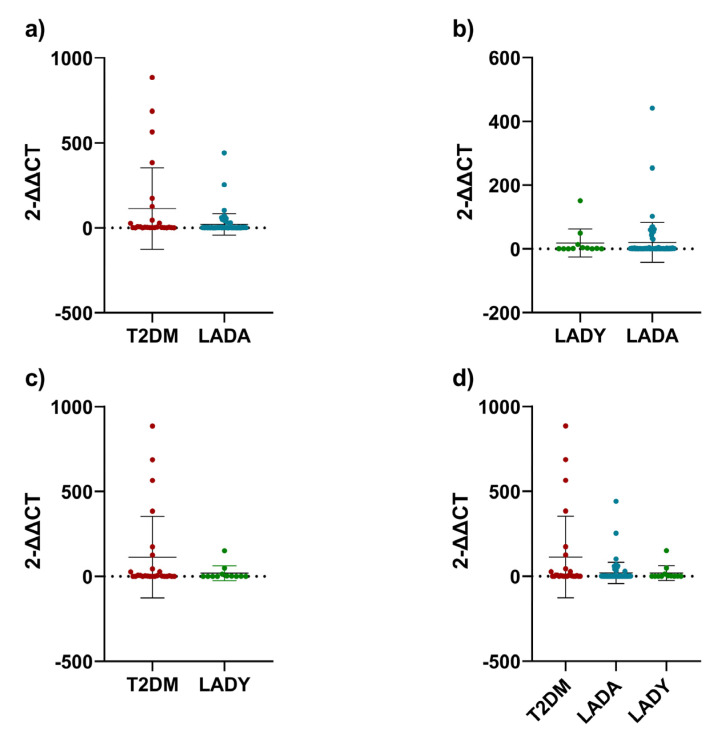
Telomere lengths. (**a**) T2DM group compared to LADA group; (**b**) LADY group compared to LADA group; (**c**) T2D group compared to LADY group; (**d**) three groups compared together. For the analysis in graphic (**a**–**c**), Mann-Whitney test was used. For graphic (**d**), Kruskal-Wallis test was used. All data points were included after adjusting the telomere length to obtain arithmetic normalization of the samples obtained from the DNA extraction and rt-PCR analysis. Error bars represent the standard deviation.

**Figure 3 biomedicines-11-00730-f003:**
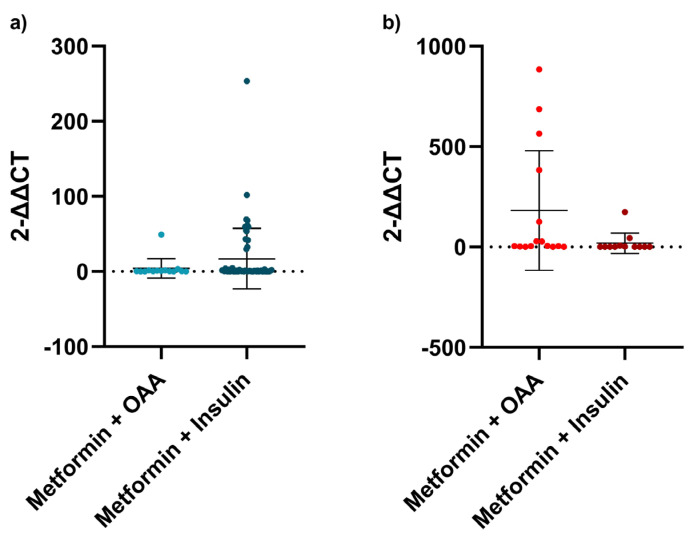
Antidiabetic agents and telomere length. (**a**) Telomere length comparison between metformin plus other antidiabetic agent and metformin plus insulin in the LADA group; (**b**) Telomere length comparison between metformin plus other antidiabetic agent and metformin plus insulin in the T2DM group. For this analysis, Mann-Whitney test was used. All patients in LADA and T2DM groups were divided according to the medical records of which antidiabetic agents were used by each patient. Error bars represent the standard deviation.

**Table 1 biomedicines-11-00730-t001:** Clinical and metabolic variables.

	LADA	LADY	T2DM	Total	*p* Value
** *n* ** **= 115 (%)**	72 (62.61)	13 (11.30)	30 (26.09)	115 (100)	
**Age (yrs)**	53.69	44.62	58.27	55.30	0.0007
**Gender**	F = 54 (75) M = 18 (25)	F = 11 (84.62) M = 2 (15.38)	F = 17 (56.67) M = 13 (43.33)	F = 82 (71.30) M = 33 (28.70)	0.0931
**BMI (kg/m^2^)**	30.01	28.05	31.32	29.79	0.0235
**HbA1c (%)**	6.46	5.64	5.28	5.79	0.8736
**Fasting plasma glucose (mg/dL)**	124.01	98.85	128.47	117.11	0.0663
**Hypertension**	29 (25.22)	6 (5.22)	17 (14.78)	52 (45.22)	0.3164
**Dyslipidemia**	19 (16.52)	1 (0.87)	8 (6.96)	28 (24.35)	0.3315
**Neuropathy**	4 (3.48)	1 (0.87)	0 (0)	5 (4.35)	0.3742
**GADA**	19.43 ± 17.28	17.88 ± 11.09	2.60 ± 1.05	17.92 ± 16.64	0.05

**Table 2 biomedicines-11-00730-t002:** Antidiabetic agents used by subjects.

	LADA	LADY	T2D	Total (%)	*p* Value
**Metformin**	68	13	29	110 (95.65)	0.6320
**Glibenclamide**	29	4	12	45 (39.13)	0.8470
**DPP-4 Inhibitors**	14	3	12	29 (25.22)	0.0916
**Insulin**	58	12	14	84 (73.04)	0.0005

DPP-4 Inhibitors: dipeptidyl peptidase 4 inhibitors.

**Table 3 biomedicines-11-00730-t003:** Correlation analysis between telomere length and clinical variables.

Characteristic	*r* Value	*p* Value
**Age**	0.02665	0.7774
**Body mass index**	0.1111	0.2370
**HbA1c**	0.1189	0.2842

**Table 4 biomedicines-11-00730-t004:** Correlation analysis between telomere length and clinical variables in the LADA group.

Characteristic	*p* Value
**Age**	0.0001
**Age of diagnosis**	0.0001

## Data Availability

The researchers interested in this study can contact the corresponding author to obtain the data.

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
