# Peer review of "Telomere Shortening in Three Diabetes Mellitus Types in a Mexican Sample"

_biomedicines, 2023, doi:10.3390/biomedicines11030730_

Round 1
Reviewer 1 Report
Even if implied, I would ask the Authors to specify that each T2D subject has been investigated for anti-beta cell autoantibodies
Author Response
Q: Even if implied, I would ask the Authors to specify that each T2D subject has been investigated for anti-beta cell autoantibodies.
A: Glutamic Acid Decarboxylase Antibodies (GADA) which were measured in all patients. GADA was used to determine the type of diabetes of each patient with the use of Human Anti-Glutamic Acid Decarboxylase Antibodies ELISA Kit following the manufacturer’s instructions. Patients with GADA >5 IU/ml and >30 years old, were diagnosed as LADA, those with GADA >5 IU/ml and <30 years old were diagnosed as LADY, and those with GADA <5 IU/ml as T2DM. These values are now included in table 1.
Reviewer 2 Report
The manuscript describes the telomere length of leucocytes from three different subtypes of diabetes (LADA; LADY, T2DM).
The writing of beta cells should be checked.
LADA is the greatest group in this analysis. Is there any correlation between age, disease onset and telomere length because LADA often is diagnosed first as a T2DM.
Are there references showing that the telomere length of leucocytes correlate with other parenchymal cells, especially beta cells in the pancreas
The authors should chexk the values in total f.e. HbA1c in Table 1.
Why the authors do not used age-matched controls for the basal values?
Author Response
Q: The writing of beta cells should be checked.
A: We are sorry, but beta cells are only written once as the beta greek symbol.
Q: LADA is the greatest group in this analysis. Is there any correlation between age, disease onset and telomere length because LADA often is diagnosed first as a T2DM.
A: Yes. Between Age and TL there is a correlation with a p <0.0001, and between disease onset and TL with a p <0.0001. Between the three variables there is a p <0.0001. These results are now included as Table 4.
Q: Are there references showing that the telomere length of leucocytes correlate with other parenchymal cells, especially beta cells in the pancreas.
A: Yes, there are some articles about that. A few references about are the following:
- https://doi.org/10.1210/jc.2014-1222
- https://doi.org/10.1111/ggi.12738
- https://doi.org/10.1038/s41598-022-08058-7
This last one may explain why our LADA and LADY patients had no major impact by their autoimmunity condition.
Q: The authors should chexk the values in total f.e. HbA1c in Table 1.
A: Done.
Q: Why the authors do not used age-matched controls for the basal values.
A: Glutamic Acid Decarboxylase Antibodies (GADA) which were measured in all patients. GADA was used to determine the type of diabetes of each patient with the use of Human Anti-Glutamic Acid Decarboxylase Antibodies ELISA Kit following the manufacturer’s instructions. Patients with GADA >5 IU/ml and >30 years old, were diagnosed as LADA, those with GADA >5 IU/ml and <30 years old were diagnosed as LADY, and those with GADA <5 IU/ml as T2DM. These values are now included in table 1.
Reviewer 3 Report
The manuscript investigated the relationship between telomere shortening and three diabetes types (LADA, LADY and T2D) in a Mexican population sample. Overall, the manuscript is well-written and the topic is interesting in the field. However, the manuscript needs some improvement for publication.
Specific comments:
1. For figure 2 and 3, please plot bar graphs with individual data points. Please also include error bars and brackets to indicate statistical significance in the graphs.
2. Please revise figure legends in the manuscript. For Figure 1, what does the Y value on the graph represent? For Figure 2 and 3, please provide more details, such as how many data points were included in each group shown in the graphs? Was any data point excluded in the graph, and why? And what statistical test was performed in each graph?
3. Page 3, line 112: “To obtain the T/S relative value” - what does abbreviation T/S mean? It was not mentioned in the text.
4. Please proofread the manuscript and correct grammar mistakes (see examples below).
Page 1, line 25: “with a BMI average of 31.32 kg/m2” and Page 2, line 95: “there was a mean body mass index (BMI) of 30.26 kg/m2”– please use the right format of BMI unit.
Page 3, line 114: “SDHA as the gen control” – should be gene control.
Page 6, line 197-198: “Considering that patients from our study had access to an antibody detecting facility, nutrition counseling and constant check-ups” – the sentence seems not completed.
Author Response
Q: For figure 2 and 3, please plot bar graphs with individual data points. Please also include error bars and brackets to indicate statistical significance in the graphs.
A: Done.
Q: Please revise figure legends in the manuscript. For Figure 1, what does the Y value on the graph represent? For Figure 2 and 3, please provide more details, such as how many data points were included in each group shown in the graphs? Was any data point excluded in the graph, and why? And what statistical test was performed in each graph?
A: Done. Now the figures include the statistical test performed for each one of the graphs, and how many data points were included in consideration.
Q: 3. Page 3, line 112: “To obtain the T/S relative value” - what does abbreviation T/S mean? It was not mentioned in the text.
A: Done.
Q: Please proofread the manuscript and correct grammar mistakes (see examples below).
A: Done. The article is in consideration for professional English review.
Round 2
Reviewer 2 Report
No points remained.
Author Response
Thank you for your time.
Reviewer 3 Report
Below is my only suggestion for the revised version:
What error bars represent in figure 2 and 3 should be included in the figure legends.
Author Response
Done. The meaning of error bars are now in the figure legends.